# Computed Tomography-Navigation™ Electromagnetic System Compared to Conventional Computed Tomography Guidance for Percutaneous Lung Biopsy: A Single-Center Experience

**DOI:** 10.3390/diagnostics11091532

**Published:** 2021-08-25

**Authors:** Morgane Lanouzière, Olivier Varbédian, Olivier Chevallier, Loïc Griviau, Kévin Guillen, Romain Popoff, Serge-Ludwig Aho-Glélé, Romaric Loffroy

**Affiliations:** 1Image-Guided Therapy Center, Department of Vascular and Interventional Radiology, François-Mitterrand University Hospital, 14 Rue Paul Gaffarel, BP 77908, 21079 Dijon, France; morgane.lanouziere@chu-dijon.fr (M.L.); olivier.chevallier@chu-dijon.fr (O.C.); kevin.guillen@chu-dijon.fr (K.G.); 2Georges-François Leclerc Cancer Center, Department of Radiology, 1 Rue du Professeur Marion, 21000 Dijon, France; ovarbedian@cgfl.fr (O.V.); lgriviau@cgfl.fr (L.G.); 3Georges-François Leclerc Cancer Center, Department of Medical Physics, 1 Rue du Professeur Marion, 21000 Dijon, France; rpopoff@cgfl.fr; 4Department of Epidemiology and Biostatistics, François-Mitterrand University Hospital, 14 Rue Paul Gaffarel, BP 77908, 21079 Dijon, France; ludwig.aho@chu-dijon.fr

**Keywords:** percutaneous lung biopsy, CT-guided biopsy, navigation system, Imactis, computed tomography

## Abstract

The aim of our study was to assess the efficacy of a computed tomography (CT)-Navigation™ electromagnetic system compared to conventional CT methods for percutaneous lung biopsies (PLB). In this single-center retrospective study, data of a CT-Navigation™ system guided PLB (NAV-group) and conventional CT PLB (CT-group) performed between January 2017 and February 2020 were reviewed. The primary endpoint was the diagnostic success. Secondary endpoints were technical success, total procedure duration, number of CT acquisitions and the dose length product (DLP) during step ∆1 (from planning to initial needle placement), step ∆2 (progression to target), and the entire intervention (from planning to final control) and complications. Additional parameters were recorded, such as the lesion’s size and trajectory angles. Sixty patients were included in each group. The lesions median size and median values of the two trajectory angles were significantly lower (20 vs. 29.5 mm, *p* = 0.006) and higher in the NAV-group (15.5° and 10° vs. 6° and 1°; *p* < 0.01), respectively. Technical and diagnostic success rates were similar in both groups, respectively 95% and 93.3% in the NAV-group, and 93.3% and 91.6% in the CT-group. There was no significant difference in total procedure duration (*p* = 0.487) and total number of CT acquisitions (*p* = 0.066), but the DLP was significantly lower in the NAV-group (*p* < 0.01). There was no significant difference in complication rate. For PLB, CT-Navigation™ system is efficient and safe as compared to the conventional CT method.

## 1. Introduction

The computed tomography (CT)-guided percutaneous lung biopsy (PLB) is a well-established and standardized procedure for characterizing pulmonary nodules, and has become a routine radiology intervention. CT guidance is now widely performed thanks to high resolution and fast volume acquisition, associated with an easy access and a low cost modality. However, standard CT-guided procedures suffer from the lack of real time visualization of the needle and expose to radiation. One of the challenges in image-guided procedures is the trajectory angle to the lesion [1,2]. Although this can be set on the pre-procedure CT scan, it is influenced by the subjectivity of the spatial representation of the operator and his experience. Reaching the target lesion might be challenging when an out-of-plane trajectory is required, source of waste of time and increase of X-ray exposure, and may cause adverse events or the procedure’s failure. CT-fluoroscopy and C-arm cone-beam CT (CBCT) allow a real-time visualization of the needle advancement while reducing the intervention time, but lead to increased radiation exposure for both the patient and operator [3,4,5,6]. Laser navigation systems allow CT-guided interventions to be carried out more quickly with a lower radiation dose. However, this type of system does not have the advantage of offering real-time guidance of the needle [7,8]. In order to improve accuracy and limit X-ray exposure, a series of navigational guidance tools have been developed, based on the principle of “augmented reality” [1,9]. These new systems allow electromagnetic [2,10,11,12,13,14,15,16,17,18], or optical tracking of devices [19,20,21,22,23], that are used for interventional radiology or surgical procedures. Data from a previous patient’s scan can be used in a model to perform virtual real time navigation. These tools are already used in various surgical fields such as neurosurgery and orthopedics [10,11]. They are also in development in the field of interventional radiology, which faces a growing demand for minimally invasive techniques. In a previous study, which included 120 CT-guided interventions with different procedures in the thoracoabdominal region, the CT-Navigation™ electromagnetic system (Imactis SAS, La Tronche, France) has demonstrated a more accurate placement of the needle with fewer control acquisitions compared with conventional methods [13]. A reduction of the delivered X-ray dose was also observed with this system compared to standard CTs, but remained non-significant. To our knowledge, few studies assessed the performance of this kind of navigational systems for CT-guided percutaneous lung biopsies. The purpose of this study was to assess the efficiency of the CT-Navigation™ system compared to conventional CT methods for PLB in terms of diagnostic success, complication rate, procedure time and radiation exposure.

## 2. Materials and Methods

### 2.1. Study Population

In this single-center retrospective study, the data of CT-Navigation™ and conventional CT guided PLB that were performed between January 2017 and February 2020 were reviewed. All patients were referred to our department for suspected primary or secondary lung malignancy. From March 2018 to February 2020, all PLB were performed using the CT-Navigation™ system in our institution, i.e., for 60 patients (NAV-group). All NAV-group patients were consecutively included. Therefore, the last 60 patients who underwent CT-guided PLB before the installation of CT-Navigation™ were included in the comparator group (CT-group).

Informed consent was obtained from every subject. Our ethics committee approved the study and waived the requirement for informed patient consent in compliance with French legislation on retrospective studies of anonymized data.

### 2.2. Computed Tomography Procedures Selection

All procedures were performed under local anesthesia by either senior interventional radiologists with more than 5 years of practice or residents under the direct supervision of a senior investigator. All procedures were carried out using a 64-row CT machine (Optima CT 660S-GT200, GE Healthcare). A preliminary chest scan was obtained and the safest and shortest access route to the target lesion and the patient positioning (supine, prone or by the side) were then chosen. All biopsies were conducted using a semi-automatic 18- or 20-Gauge needle (ARGON, SuperCore™ Semi-Automatic Biopsy Instrument, Dallas, TX, USA) with a co-axial technique (ARGON, Co-axial Introducer Needle, Dallas, TX, USA). Tissue samples were fixed in a formaldehyde solution and submitted to the pathology department for histological diagnosis. A final complete chest CT scan was performed after needle removal in search of early complications.

Procedures were divided into 3 steps: step ∆1, from biopsy planning to initial needle placement, step ∆2, from skin puncture to needle progression toward the target lesion, and step ∆3, from biopsy to final control scan.

#### 2.2.1. Conventional CT-Guided Procedures: CT-Group

Regarding the CT-control group, the intervention was performed according to the conventional procedure using the following acquisition parameters: 120 kV, average of 185 mA, pitch of 0.53, slice thickness of 2.5 mm. The best needle trajectory was defined by the interventional radiologist on the initial CT-scan using the CT console. Entry point was obtained by using the CT gantry laser line to indicate the axial position. Every time the needle was advanced, a CT control acquisition was performed to assess its position, and images were checked by the operator at the CT console.

#### 2.2.2. CT-Navigation™ Guided Procedures: NAV-Group

For the CT-Navigation™-guided procedures, the following parameters were used: 120 kV, an average of 281 mA, pitch 1.38, slice thickness of 1.25 mm. The CT-Navigation™ system is composed of a station with a touch screen and an electromagnetic locator (Figure 1). This locator is made of: (1) a field generator placed on the patient’s skin near the puncture site that allows an automatic registration of magnetic and CT coordinates; (2) and a magnetic receiver located inside the needle holder. The base unit is a freestanding device placed next to the CT table. First, all DICOM images of the pre-procedure CT-scan are uploaded automatically to the navigation station within a few seconds. Thanks to the sensor unit integrated into the needle holder, the system dynamically displays the position and orientation of the needle-holder on two-perpendicular 2D reconstructed CT-images that are extracted from the acquired 3D CT volume and projected onto the workstation. The radiologist can thus use the needle holder like a 3D mouse to explore the scanner volume, locate the target and plan the entry point as well as the trajectory directly in the CT room. Once the sterile preparation is performed using the cover and the sterile needle from the CT-Navigation™ kit, the radiologist places the needle in the needle holder, stands on the entry point and inserts the needle when the displayed trajectory is optimal. Real time needle depth is indicated by entering the length of the used needle, allowing visualization of the exact distance remaining to be inserted. During the needle insertion phase, additional CT acquisition can be performed and transferred to the workstation in order to check the progression of the needle tip. A video describing step by step the use of the navigation system is available in [24].

### 2.3. Outcomes Definition

The primary endpoint was to assess the feasibility of PLB assisted by the CT-Navigation™ system, defined by diagnostic success. Diagnostic success was achieved when histological diagnosis was accomplished with the provided samples. Secondary endpoints were as follows: technical success, that was defined as the achievement of the needle progression to the targeted lesion; total intervention duration; duration of step ∆1, step ∆2 and step ∆3; number of CT acquisitions; mean radiation exposure defined by the dose length product (DLP) during step ∆1 and step ∆2, and then reported to the entire intervention; minor and major complications as defined by the Society of Interventional Radiology [25].

Additional parameters were recorded and compared between groups: patient’s age, body mass index (BMI), patient position during procedure (supine, prone, oblique), operator experience (senior or resident), lesion’s size, distance of the lesion to the skin and to the pleura, lesion’s localization across the different pulmonary lobes, the two trajectory angles on the 2D reconstructed perpendicular images in the axial plane (angle 1) and the sagittal or frontal oblique plane (angle 2). Figure 2 shows an example of angle 1 and angle 2.

In addition, every preliminary chest scan was reviewed by a blinded interventional radiologist with more than 5 years of experience (O.V.) who classified the procedures as “easy” and “difficult” for sub-group analysis.

### 2.4. Statistical Analysis

Continuous and discrete variables were described using the median as well as first and third quartile (M [Q1; Q3]). For each variable, normality was assessed with the Shapiro–Wilk test and the comparison between patient groups was accordingly carried out with a Mann–Whitney U test or a t-test. Categorical variables were analyzed with either Fisher tests or Chi-square tests. The chosen level of significance α was 0.05.

Two sub-analyses were performed pertaining to the intervention stage (step ∆1/step ∆2) and the procedure difficulty (easy/difficult). The statistical analysis was performed using R software, version 4.0.3.

## 3. Results

### 3.1. Population Characteristics

Patients and target lesion characteristics are shown in Table 1. Sixty patients were included in each group. There was no significant difference between the NAV and CT groups in the following patients’ characteristics: age (*p* = 0.576), sex ratio (*p* = 0.465), BMI (*p* = 0.576) and patient’s position (*p* = 0.187).

### 3.2. Operators Experience

A total of 10 operators participated in the procedures (8 seniors, 2 residents). There was no significant difference in operator experience (senior/resident) between the two groups (*p* = 0.328).

### 3.3. Lesions Characteristics

Lesion size was significantly lower in the NAV-group with a median lesion size of 20 mm (range, 5–90 mm) versus 29.5 mm (range, 6–140 mm) in the CT-group (*p* = 0.007). Significantly more biopsies of lesions < 2 cm were performed in the NAV-group (29/60) than in the CT-group (14/60) (*p* = 0.007). There was no significant differences in lesion localization across the different lung lobes (*p* = 0.754), and distance of the lesion to the pleura (*p* = 0.340) between the two groups. The distance of the lesion to the skin was significantly larger in the NAV-group (*p* = 0.028). Significantly more lesions were classified as “difficult” in the NAV-group (27/60) than in the CT-group (11/60) (*p* = 0.0017). Trajectory angles 1 and 2 were significantly larger in the NAV-group (*p* < 0.001 and *p* < 0.001, respectively).

### 3.4. Technical and Diagnostic Success Rates

Technical and diagnostic success results are shown in Table 2. Diagnostic success results for “easy” and “difficult” lesion subgroups are shown in Table 3 and Table 4. Technical success and diagnostic success were, respectively, achieved in 57 (95.0%) and 56 (93.3%) patients in the NAV-group and in 56 (93.3%) and 55 (91.6%) in the CT-group, with no significant difference between the two groups (*p* = 1.00 and *p* = 1.00, respectively). For “easy” and “difficult” lesions, the diagnostic success rates were higher in the NAV-group (100.0% and 85.2%, respectively) than in the CT-group (95.9% and 72.7%, respectively), with no significant difference (*p* = 0.513 and *p* = 0.390, respectively).

### 3.5. Procedure Duration and Radiation Dose Exposure

Procedure duration, number of CT acquisitions and radiation dose are shown in Table 2. For “easy” and “difficult” lesion subgroups, these results are shown in Table 3 and Table 4.

#### 3.5.1. Step ∆1

The duration of step ∆1 was significantly longer in the NAV-group (13 min [11; 17]) than in the CT-group (11 min [8; 15]) (*p* = 0.018), with no significant difference for “easy” and “difficult” lesion subgroups (*p* = 0.079 and *p* = 0.701, respectively). The DLP was significantly lower in the NAV-group (142.9 mGy.cm (105.9; 219.5)) as compared to the CT-group (188.2 mGy.cm (123.4; 360.7); *p* = 0.013). There was no significant difference regarding the number of CT acquisitions during step ∆1. The acquisition length of the registration CT-scan was significantly shorter in the NAV-group (143.8 mm (111.9; 205.6)) than in the CT-group (264.4 mm (166.9; 350); *p* < 0.001). For easy lesions, there was no significant difference in step ∆1 duration. However, the number of CT acquisitions and the DLP were similar between groups and significantly lower in the NAV-group, respectively (*p* = 1.000 and *p* = 0.011, respectively). For “difficult” lesions, there was no significant difference in the step ∆1 duration, number of CT acquisitions and DLP (*p* = 0.701, *p* = 0.869, and *p* = 1.000, respectively).

#### 3.5.2. Step ∆2

The step ∆2 duration was significantly shorter in the NAV-group than in the CT-group (*p* = 0.042). DLP was significantly lower in the NAV-group (*p* = 0.001), with the number of acquisitions being similar (*p* = 0.053). However, the acquisition length of each control acquisition during the procedure was significantly longer in the NAV-group than in the CT-group (594.4 mm (365.6; 780) vs. 420 mm (312.5; 556.9); *p* = 0.001). For “easy” lesions, step ∆2 duration, the number of CT acquisitions and the DLP were significantly lower in the NAV-group (*p* = 0.015, *p* < 0.001, and *p* < 0.001, respectively). For “difficult” lesions, there was no significant difference in the step ∆2 duration, number of CT acquisitions and DLP (*p* = 0.206, *p* = 0.459, and *p* = 0.158, respectively).

#### 3.5.3. Whole Procedure

In total, there was no significant difference in whole procedure duration or in the number of control acquisitions (*p* = 0.497 and *p* = 0.066, respectively). However, the total radiation dose was significantly lower (*p* < 0.01) in the NAV-group than in the CT-group (1059 mGy.cm (747; 1456) versus 1481 mGy.cm (1255.5; 1873); *p*< 0.0001).

### 3.6. Complications

No major complication occurred in NAV-group and 3 major complications (5.0%) occurred in CT-group, with no significant difference (*p* = 0.244). Complications are detailed in Table 5.

## 4. Discussion

With a population of 120 patients, the present study based on our initial experience with the CT-Navigation™ electromagnetic system demonstrates that the use of this system for PLB is efficient and safe compared to conventional CT methods. The technical and diagnostic success rates were very high (95.0% and 93.3%, respectively) with no significant difference when compared to the conventional CT method. The whole procedure duration was very similar between the two groups, with no significant difference. The total radiation dose exposure was significantly lower in the NAV-group (*p* < 0.0001).

Our results are consistent with previous reports in terms of diagnostic success for PLB [20]. In a study, which included 180 patients, Grasso et al. found a diagnostic success rate of 96% in the navigation group and 90% in the standard group [20].

Despite no significant difference being found in the total procedure duration between the two groups, the duration of step ∆2, that corresponds to the time duration from the puncture to needle progression toward the targeted lesion, was significantly shorter in the NAV-group. This procedure stage is the most uncomfortable for the patient. The navigation system offers the ability to find the best trajectory quickly, decreasing the need for needle repositioning. In the same way as in the Durand et al. study, preparation time (step ∆1) was longer in the NAV-group as compared to the CT-group [13]. It may be explained by the time required to install the device. The installation of the navigation system requires to be fully integrated into the workflow. The operators and the imaging technologists must be accustomed to the use of the equipment, with a real learning curve over time. In our study, the longer preparation time may also be linked to the number of radiologists and residents who participated in the interventions. Some used the navigation system only once before this study, which may increase the average preparation time. Additionally, the system enables us to explore many different trajectories in the 3D volume, allowing the radiologist to choose the best option, that may lengthen the preparation time. Although a previous study showed that using the electromagnetic guidance minimized the impact of operator experience [15], allowing unskilled operators to obtain as good results as the best trained interventional radiologists, the system can still slow down the procedure for the most trained operators and for simple procedures [1].

In our study, less intermediate control acquisitions were required in the NAV-group as compared to the CT-group. Nevertheless, this difference did not reach the level of statistical significance. In a prospective study comparing the CT-Navigation™ system to the conventional method, Durand et al. showed a significant decrease in the number of control acquisitions [13]. However, our study focused only on percutaneous lung biopsies. The lack of synchronization with the respiratory movements could explain the number of intermediate control. Although these movements can be critical for the navigation systems as well as for the conventional method, the navigation system superimposes the supposed path of the needle on previously acquired images. In this way, every movement of the patient between the scan previously acquired and needle placement can lead to a path error. This leads to a greater number of controls and therefore an increase in the radiation dose and procedure time. This is particularly true for small lung lesions, but also lesions localized in the lower lobes, more sensitive to respiratory motions. Studies have shown that this kind of drawback can possibly be overcome by the use of fusion systems with ultrasound images [1,14,26]. However, this kind of system is not applicable in the specific field of lung biopsies. Despite these respiratory motions, the use of CT-Navigation™ system was reliable enough to demonstrate a very high diagnostic success rate. Note that the number of CT acquisitions was significantly lower in the NAV-group during step ∆2 for “easy” lesions, meaning that the radiologist’s confidence in this system is probably higher for this kind of lesion.

Our study found significant lower radiation exposure using the navigation system than using the conventional method. However, these results must be balanced by the fact that the CT acquisition parameters for lung biopsies were modified during the study. When the CT-Navigation™ system was implemented in our department, the CT acquisition parameters for lung biopsies were optimized. The difference in acquisition parameters may thus partly explain the difference in radiation dose between the two groups. However, the radiation doses in each group were comparable to those presented in the literature [19]. For “easy” lesions, less CT acquisitions were required during step ∆2 in the NAV-group. A dose reduction is thus expected when using CT-Navigation™ system, at least for this kind of lesion.

CT-Navigation™ provides a reliable tool to assist PLB procedures even for small lesions. Lesions included in the NAV-group were significantly smaller. In addition, more complex trajectories were performed in the NAV-group, with significantly greater angulations than in the CT-group. Significantly more lesions were classified “difficult” in the NAV-group. These data may suggest that the physician feels more confident with the CT-Navigation™ system when faced with smaller lesions or lesions that are more difficult to access with an out-of-plane trajectory. The use of CT-Navigation™ systems might thus contribute to the achievement of more complex PLB procedures. A previous phantom study evaluating the CT-Navigation™ system showed that operators, including novice ones, performed faster and more accurately punctures with out-of-plane trajectories with the help of the navigation system [16]. In accordance with more out-of-plane trajectories when using the navigation system, the acquisition length was longer in the CT-NAV group than in the CT-group during step ∆2.

All the identified complications were related to the usual adverse events of the PLB procedure. None of these adverse events could be directly related to the use of the navigation system. No difference was found between the two groups. The rates of pneumothorax (31.6% in the CT-group; 25% in the NAV-group; *p* = 0.418) were similar to those found in the literature [27,28].

The CT-Navigation™ system had already been studied for various procedures with a high rate of technical success [13,14,18,29], but the present study is the first study evaluating this system in the specific field of lung biopsies. An experiment similar to ours has been described by Grand et al. comparing lung biopsies with CT-fluoroscopy guidance or with CT-fluoroscopy using an electromagnetic navigation system in 60 patients [12]. No significant difference was found in terms of procedure time, radiation dose, number of control acquisitions and complication rate. However, eight cases required the abandonment of the navigation system due to technical issue. In the present study, the navigation system has proven to be reliable. No dysfunction has been reported. The automatic recording and detection of the magnetic transmitter on the patient’s skin has always been effective and has never prevented the use of the navigation system. In a randomized study on 180 patients using a navigation system based on optical tracking in the field of lung biopsies, Grasso et al. found a significant reduction in the procedure time, the number of control scans and the radiation dose [20]. In another study, this optical navigation system called SIRIO (MASMEC S.p.A., Modugno, BA, Italy) has been shown to be more accurate for percutaneous lung biopsies of lesions <20 mm [21]. With this navigation system, Ianelli et al. showed better efficiency in terms of dose reduction, procedural time and diagnostic success for lesions <10 mm compared to the conventional method, with fewer post-procedural complications [19]. These optical and electromagnetic navigation systems both have their advantages and disadvantages [30]. The optical-based systems use fiducial markers located on the instrument and on the patient’s skin that are detected by stereoscopic cameras, which requires a direct line of view between the cameras and the fiducials and may limit their application in clinical practice. Although electromagnetic guidance systems do not suffer from these kind of constraints, they can be affected by magnetic susceptibility artifacts. One of the advantages of the SIRIO optical navigation system is the synchronization to the patient’s respiratory movements, which is particularly interesting for lung biopsies and may partly explain the significant decrease in the number of control acquisitions, radiation dose and procedure time compared to the standard method.

To our knowledge, six studies about the use of the CT-Navigation™ system on patients have been published [13,14,18,29,31,32]. Mavrovi et al.’s retrospective study with 12 patients, the CT-Navigation™ system has been used as a guidance system for radiological percutaneous osteosynthesis and cementoplasty of malignant pathological fracture of the proximal femur [31]. Although the study’s main subject was not focused on the navigational system, an excellent technical success of 100% was found [31]. In Moulin et al.’s study, the feasibility of percutaneous fixation by internal cemented screw for the prevention or palliation of pelvic or femoral neck fractures using the CT-Navigation™ system was evaluated [18]. With a total of 76 screws inserted, technical success was achieved in 96% (48/50 patients), with high accuracy even when a significant craniocaudal angulation was needed. With a total of 120 enrolled patients, a prospective randomized trial assessed the accuracy and usability of this system by comparing conventional and navigated procedures in a full range of routine CT interventions on the chest, abdomen, pelvis and bones, such as biopsy, drainage, tumor ablation, sympathicolysis and joint infiltration [13]. A significant improvement of the accuracy was found with the use of the system, with a gain in accuracy remaining significant in both the easy and difficult intervention subgroups and in a subgroup analysis based on operator experience [13]. The retrospective study of Teriitehau et al. focused on the radiation doses delivered to patients during percutaneous vertebroplasty [29]. With 15 consecutive patients who underwent a conventional CT guided procedure, and 22 patients who underwent a procedure with the use of the CT-Navigation™ system, a significant reduction in the DLP by a 3.2 factor was found. The use of this navigation system also demonstrated an important reduction of the procedure duration (50%) [29]. Two studies focused on the use of the CT-Navigation™ system for hepatic tumor ablation [14,32]. In Volpi et al.’s study, 27 percutaneous ablations of small hepatic tumors (<2 cm) that were invisible on ultrasounds and difficult to reach on CT, requiring a double-oblique approach, were performed using both CT-Navigation™ system and high-frequency jet-ventilation (HFJT) [14]. Complete ablation was obtained at the six-month follow-up, with the ablation probe correctly placed on the first pass in 96%. This last study suggests an excellent accuracy offered by the CT-Navigation™ system when combined with (HFJT). In a more recent study, the use the CT-Navigation™ system for microwave ablation of hepatic tumors was also investigated [32]. Two groups of 17 patients were compared. The application of the CT-Navigation™ system was feasible in 14 cases (82%). The mean total deviation of the antenna feed point was significantly lower in the navigation group as well as the mean number of control scans and the patient radiation exposure [32]. All these studies support the use of the CT-Navigation™ system, particularly for improving the targeting accuracy.

Our study has several limitations. First, it is a retrospective and non-randomized study conducted in a single center. Second, different acquisition parameters were used in the NAV-group and in the CT-group, which may explain the significant lower radiation dose found in the NAV-group. Nonetheless, the CT-Navigation™ system did not lead to an increase in the radiation dose despite carrying out more complex procedures with smaller lesions and out-of-plane trajectories, requiring longer acquisition lengths. Last, this study is based on the first two years’ experience with the CT-Navigation™ system in our center, with many different operators, more or less experienced in PLB and in the use of this navigation system. A progression curve over time and a possible improvement of these results are expected.

## 5. Conclusions

The CT-Navigation™ electromagnetic system is efficient, reliable and safe for PLB, compared to conventional CT method. However, in the particular field of PLB, some improvements, such as synchronization with respiratory motions, would be beneficial, even if the use of this system without synchronization already demonstrated a very high diagnostic success rate. A prospective randomized controlled study with a larger cohort is needed to confirm the real impact of the CT-Navigation™ system on the radiation dose, number of control acquisitions and procedural time for CT-guided lung biopsies.

## Figures and Tables

**Figure 1 diagnostics-11-01532-f001:**
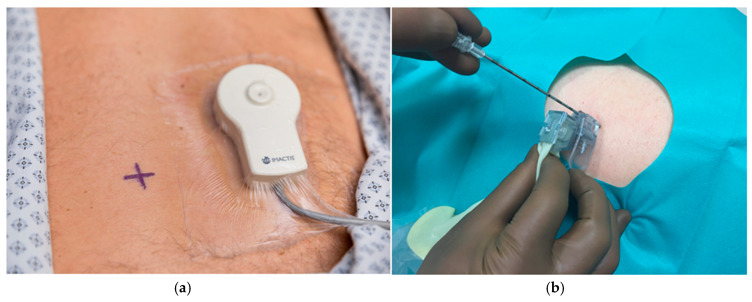
(**a**) Field generator placed on the patient’s skin; (**b**) magnetic receiver located inside the needle holder. (**c**) The station dynamically displays the position and orientation of the needle-holder on two-perpendicular 2D reconstructed CT-images.

**Figure 2 diagnostics-11-01532-f002:**
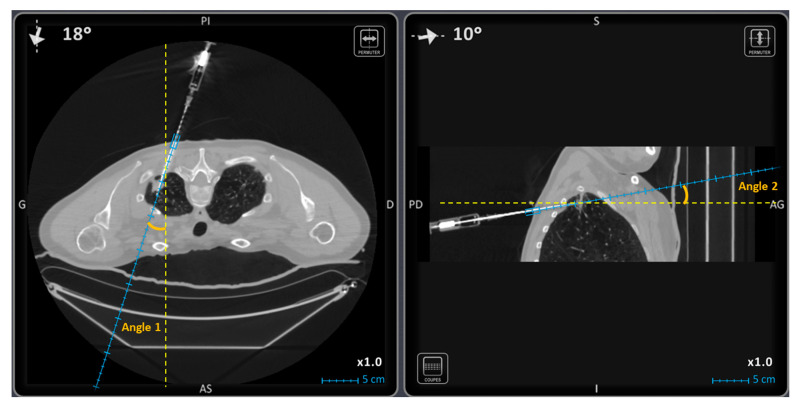
The two trajectory angles on the 2D reconstructed perpendicular images in the axial plane on the **left** (angle 1) and the sagittal oblique plane on the **right** (angle 2).

**Table 1 diagnostics-11-01532-t001:** Characteristics of the patients and the lesions.

	NAV Group	CT Group	*p*
No. of patients	60	60	
Sex ratio M/F	28/32	33/27	0.465
Mean age (range) (years)	66 (58; 72)	68 (60; 73)	0.576
BMI (range) (kg.m^−2^)	26 (23; 28)	24 (21; 29)	0.187
Position (P/SU/SI)	25/33/2	33/26/1	0.187
Operator			
Senior (%)	57 (95.0)	54 (90.0)	0.328
Resident (%)	3 (5.0)	6 (10.0)	
Lesion size (mm)	20.0 [12.8; 37.0]	29.5 [20.0; 49.5]	0.007
No. of lesions < 2 cm (%)	29 (48.3)	14 (23.3)	0.007
Distance to the skin ± SD (mm)	70.2 ± 22.1	62.2 ± 16.8	0.028
Distance to pleura (mm)	18 [10.0; 33.3]	21.5 [7.0; 28.0]	0.340
Localization (%)			0.754
Right lower lobe	12 (20.0)	13 (21.7)	
Right upper lobe	23 (38.3)	19 (31.7)	
Middle lobe	3 (5.0)	4 (6.7)	
Left lower lobe	10 (16.7)	7 (11.7)	
Left upper lobe	12 (20.0)	17 (28.3)	
No. of easy lesions (%)	33 (55.0)	49 (81.7)	0.002
No. of difficult lesions (%)	27 (45.0)	11 (18.3)	
Trajectory angle 1 (°)	15.5 [3.0; 31.5]	6.0 [2.0; 13.0]	0.0008
Trajectory angle 2 (°)	10.0 [3.0; 16.3]	1.0 [1.0; 5.0]	<0.0001

No., number; M, male; F, female; BMI, body mass index; P, prone position; SU, supine position; SI, by the side position; mm, millimeters; %, percentage. Continuous and discrete variables are presented using median, first and third quartile (M [Q1; Q3]) unless otherwise specified. *p* < 0.05 was considered as significant.

**Table 2 diagnostics-11-01532-t002:** Technical and diagnostic success, procedure duration, number of CT acquisitions and radiation dose.

	NAV Group	CT Group	*p*
Technical success (%)	57 (95.0)	56 (93.3)	1.000
Diagnostic success (%)	56 (93.3)	55 (91.6)	1.000
Step ∆1			
Duration (min)	13 [11; 17]	11 [8; 15]	0.018
No. of CT acquisitions	1 [1; 1]	1 [1; 1]	1.000
DLP (mGy.cm)	142.9 [105.9; 219.5]	188.2 [123.4; 360.7]	0.013
Step ∆2			
Duration (min)	9 [7; 12]	12 [9; 15]	0.042
No. of CT acquisitions	6 [4; 8]	7 [5; 9]	0.053
DLP (mGy.cm)	611.6 [415.2; 888.7]	849.5 [574.4; 1089.5]	0.001
Whole procedure:			
Duration (min)	28 [25; 35]	29 [23; 33]	0.497
No. of CT acquisitions	8 [7; 10]	9 [8; 11]	0.066
DLP (mGy.cm)	1059.3 [747.0; 1456.0]	1481.3 [1255.5; 1872.9]	<0.0001

No., number; %, percentage; DLP, dose length product; min, minute. Continuous and discrete variables are presented using median, first and third quartile (M [Q1; Q3]) unless otherwise specified. *p* < 0.05 was considered as significant.

**Table 3 diagnostics-11-01532-t003:** Outcomes for “easy” lesions in terms of diagnostic success, procedure duration, number of CT acquisitions and radiation dose.

	NAV Group	CT Group	*p*
No. of “easy” lesions (%)	33 (55)	49 (81.7)	
Diagnostic success (%)	33 (100.0)	47 (95.9)	0.513
Step ∆1			
Duration (min)	12 [11; 15]	11 [8; 13]	0.079
No. of CT acquisitions	1 [1; 1]	1 [1; 1]	1.000
DLP (mGy.cm)	138.3 [96.5; 183.3]	227.7 [123.4; 392.3]	0.011
Step ∆2			
Duration (min)	9 [7; 10]	11 [8; 15]	0.015
No. of CT acquisitions	5 [4; 6]	7 [5; 9]	<0.001
DLP (mGy.cm)	475.1 [322.4; 713.9]	833.4 [548.9; 1058.2]	<0.001

No., number; DLP, dose length product; min, minute. Continuous and discrete variables are presented using median, first and third quartile (M [Q1; Q3]) unless otherwise specified. *p* < 0.05 was considered as significant.

**Table 4 diagnostics-11-01532-t004:** Outcomes for “difficult” lesions in terms of diagnostic success, procedure duration, number of CT acquisitions and radiation dose.

	NAV Group	CT Group	*p*
No. of difficult lesions (%)	27 (45)	11 (18.3)	
Diagnostic success (%)	23 (85.2)	8 (72.7)	0.390
Step ∆1			
Duration (min)	14 [12; 19]	14 [10; 19]	0.701
No. of CT acquisitions	1 [1; 1]	1 [1; 1]	1.000
DLP (mGy.cm)	148.6 [112.5; 230.6]	165.3 [138.0; 173.5]	1
Step ∆2			
Duration (min)	11 [9; 15]	13 [12; 15]	0.206
No. of CT acquisitions	8 [6; 9]	9 [7; 10]	0.459
DLP (mGy.cm)	821.7 [465.7; 1021.9]	1085.9 [867.4; 1198.6]	0.158

No., number; DLP, dose length product; min, minute. Continuous and discrete variables are presented using median, first and third quartile (M [Q1; Q3]) unless otherwise specified. *p* < 0.05 was considered as significant.

**Table 5 diagnostics-11-01532-t005:** Peri-procedural complications.

	NAV Group	CT Group	*p*
No. of procedures	60	60	
No. of complications (%)			
Pneumothorax	15 (25.0)	19 (31.6)	0.418
Pneumothorax requiring chest tube	0 (0.0)	3 (5.0)	0.244
Thoracic wall hematoma	0 (0.0)	0 (0.0)	-
Hemothorax	0 (0.0)	0 (0.0)	-
Intra-alveolar hemorrhage	21 (35.0)	16 (26.7)	0.323
Hemoptysis moderate or severe	0 (0.0)	0 (0.0)	-
Systemic air embolism	0 (0.0)	0 (0.0)	-

No., number; %, percentage; *p* < 0.05 was considered as significant.

## Data Availability

All the study data are reported in this article.

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
