# Peer review of "Computed Tomography-Navigation™ Electromagnetic System Compared to Conventional Computed Tomography Guidance for Percutaneous Lung Biopsy: A Single-Center Experience"

_diagnostics, 2021, doi:10.3390/diagnostics11091532_

Round 1
Reviewer 1 Report
Comments:
EM tracking in medicine as part of computer-assisted interventions (CAI) is already an important part of modern diagnostics and patient care. Therefore such publications on this topic contribute to the understanding of the various applications in medicine.
The authors report a study of the efficacy of computed tomography (CT)-Navigation™ electromagnetic system compared to conventional CT method for percutaneous lung biopsies (PLB), two figures (a-c), five Tables and 30 references.
Based on 120 patients, the study shows that the CT Navigation electromagnetic system used for PLB is comparatively safe as conventional CT method, it enables to explore many different trajectories in the 3D volume to find the best trajectory in the 3D volume, and is also useful in the case of small lesions. The obtained results are encouraging and obviously show a success of the applied method.
The introduction summarizes recent publications in the field of computer-assisted interventions (CAI), i.e., here EM tracking in medicine. Then in the next chapter all working methods, such as operators, CT equipment, semi-automatic biopsy instruments, etc., are carefully described and illustrated.
The results are presented in five tables comparing the NAV and CT groups. They are all described in detail; noteworthy is the detailed listing of peri-procedural complications.
In the discussion, the authors summarize the results of the study and point out the advantages of CT-guided percutaneous lung biopsy (PLB) (significantly lower radiation exposure) compared with the classic CT method.
The figures are clear and well presented, clearly and neatly described, all tables are understandable, the references seem complete.
Author Response
Response to Reviewer 1 Comments
EM tracking in medicine as part of computer-assisted interventions (CAI) is already an important part of modern diagnostics and patient care. Therefore such publications on this topic contribute to the understanding of the various applications in medicine.
The authors report a study of the efficacy of computed tomography (CT)-Navigation™ electromagnetic system compared to conventional CT method for percutaneous lung biopsies (PLB), two figures (a-c), five Tables and 30 references.
Based on 120 patients, the study shows that the CT Navigation electromagnetic system used for PLB is comparatively safe as conventional CT method, it enables to explore many different trajectories in the 3D volume to find the best trajectory in the 3D volume, and is also useful in the case of small lesions. The obtained results are encouraging and obviously show a success of the applied method.
The introduction summarizes recent publications in the field of computer-assisted interventions (CAI), i.e., here EM tracking in medicine. Then in the next chapter all working methods, such as operators, CT equipment, semi-automatic biopsy instruments, etc., are carefully described and illustrated.
The results are presented in five tables comparing the NAV and CT groups. They are all described in detail; noteworthy is the detailed listing of peri-procedural complications.
In the discussion, the authors summarize the results of the study and point out the advantages of CT-guided percutaneous lung biopsy (PLB) (significantly lower radiation exposure) compared with the classic CT method.
The figures are clear and well presented, clearly and neatly described, all tables are understandable, the references seem complete.
Reply: Thank you very much for your comments. Nothing has been added.
Reviewer 2 Report
I have read this manuscript in great details and I noticed that the manuscript has a lot of new content.
This study firstly discusses the use of the CT-Navigation™ system in image-guided procedures in the specific field of lung biopsies. The results demonstrate a very high diagnostic success rate and a significantly lower radiation exposure than compared to using the conventional method. Also, the CT-Navigation™ system is a reliable tool to assist percutaneous lung biopsy procedures for small lesions. The authors claimed that the CT-Navigation™ system could be operated even by novice operators and produce accurate results.
In order to have a better form of the manuscript, a section devoted to the literature review should be added. In my opinion, the literature review is presented to show existing works related to this topic and to bring out the gap in knowledge. Authors have shown few existing works; thus, they need to provide more details on the topic.
The results are very encouraging and this paper is definitely publishable after these improvements.
Author Response
Response to Reviewer 2 Comments
I have read this manuscript in great details and I noticed that the manuscript has a lot of new content.
This study firstly discusses the use of the CT-Navigation™ system in image-guided procedures in the specific field of lung biopsies. The results demonstrate a very high diagnostic success rate and a significantly lower radiation exposure than compared to using the conventional method. Also, the CT-Navigation™ system is a reliable tool to assist percutaneous lung biopsy procedures for small lesions. The authors claimed that the CT-Navigation™ system could be operated even by novice operators and produce accurate results.
In order to have a better form of the manuscript, a section devoted to the literature review should be added. In my opinion, the literature review is presented to show existing works related to this topic and to bring out the gap in knowledge. Authors have shown few existing works; thus, they need to provide more details on the topic.
The results are very encouraging and this paper is definitely publishable after these improvements.
Reply: Thank you very much for your comment. So far, few studies have indeed focused on the use of the CT-Navigation™ system. Almost all papers have already been cited in the introduction. As per your recommendation, we added a new paragraph describing the main results of the CT-Navigation™ system studies in the discussion section. We have decided to add it at the end of the discussion section, since the main topic of our study was lung biopsy puncture and these articles mainly deal with other target organs or with other types of procedures. This new paragraph should improve the knowledge on this topic as suggested.